# Conservation of uORF repressiveness and sequence features in mouse, human and zebrafish

Guo-Liang Chew[1,†], Andrea Pauli[1,†] & Alexander F. Schier[1,2,3,4,5]

Upstream open reading frames (uORFs) are ubiquitous repressive genetic elements in vertebrate mRNAs. While much is known about the regulation of individual genes by their uORFs, the range of uORF-mediated translational repression in vertebrate genomes is largely unexplored. Moreover, it is unclear whether the repressive effects of uORFs are conserved across species. To address these questions, we analyse transcript sequences and ribosome profiling data from human, mouse and zebrafish. We find that uORFs are depleted near coding sequences (CDSes) and have initiation contexts that diminish their translation. Linear modelling reveals that sequence features at both uORFs and CDSes modulate the translation of CDSes. Moreover, the ratio of translation over 5' leaders and CDSes is conserved between human and mouse, and correlates with the number of uORFs. These observations suggest that the prevalence of vertebrate uORFs may be explained by their conserved role in repressing CDS translation.

[1] Department of Molecular and Cellular Biology, Harvard University, Cambridge, Massachusetts 02138, USA. [2] The Broad Institute of Massachusetts Institute of Technology and Harvard, Cambridge, Massachusetts 02142, USA. [3] FAS Center for Systems Biology, Harvard University, Cambridge, Massachusetts 02138, USA. [4] Center for Brain Science, Harvard University, Cambridge, Massachusetts 02138, USA. [5] Harvard Stem Cell Institute, Harvard University, Cambridge, Massachusetts 02138, USA. † Present addresses: Basic Sciences Division, Fred Hutchinson Cancer Research Center, Seattle, Washington 98109, USA (G.-L.C.); Research Institute of Molecular Pathology (IMP), 1030 Vienna, Austria (A.P.). Correspondence and requests for materials should be addressed to G.-L.C. (email: chewgl@fredhutch.org).

Ribosomal preinitiation complexes (PICs) typically scan across the 5′ leaders (also known as 5′ untranslated regions or 5′ UTRs) of eukaryotic mRNAs before initiating translation at the start codon of coding sequences (CDSes)[1,2]. Open reading frames (ORFs), as defined by a start codon and a downstream in-frame stop codon, can occur upstream of CDSes in 5′ leaders; many of these upstream open reading frames (uORFs) have been found to be repressive, presumably because translation of uORFs can occur at the expense of translation of downstream CDSes[3–5].

Despite their repressive effects, uORFs are prevalent in vertebrate transcriptomes (present in ∼50% of human and mouse messenger RNAs (mRNAs) and in ∼65% of zebrafish mRNAs)[6–9], and many vertebrate uORFs are translated, as evidenced by ribosome profiling[9–16] and mass spectrometry[14,17–20]. It has not been explored, however, how broadly uORFs repress the translation of vertebrate coding sequences. Moreover, it is unclear whether and how the regulatory relationships between uORFs and CDSes are conserved[16].

Here we address these questions by analysing uORF repressiveness in human, mouse and zebrafish, using three independently generated ribosome profiling data sets[9–11]. By taking advantage of the nucleotide resolution and quantitative nature of ribosome profiling data[21], we quantify the range and conservation of uORF-mediated translational repression and determine how various transcript features modulate uORF repressiveness and CDS translation efficiency (TE). Our analyses suggest that while the repressiveness and sequence features of uORFs are conserved in vertebrates, CDS translation is modulated by the combined effects of various transcript sequence features.

## Results

**Study design.** Previous studies have identified sequence features that modulate the repressive effects of uORFs on the translation of CDSes: the sequence and secondary structure around uORF starts (initiation context) influence the efficiency of translation initiation at uORFs[2], while the distance between a uORF and CDS affects the efficiency of reinitiation following translation of a uORF[3,22]. We used these well-established features to analyse the repressive potential of human, mouse and zebrafish uORFs. uORFs were defined as ATG-Stop delimited sequences beginning upstream of the CDS start (see Methods for details). Unless otherwise stated, results discussed in main figures and text are for mouse ES cell ribosome profiling data[10]; similar results were observed in the analyses of zebrafish and human data, and are provided in Supplementary Materials.

**uORF initiation context sequence.** To define the sequence motifs that promote translational initiation, we constructed weighted position-specific scoring matrices (PSSMs) from the initiation contexts of CDSes (±10 nucleotides around AUG start codons). As a training set, we used CDS initiation contexts of annotated protein-coding mRNAs lacking uORFs and weighted their contribution using TE values (density of ribosome profiling reads over CDS normalized by transcript expression levels; see Methods) (Fig. 1a). These PSSMs were subsequently used to score individual initiation contexts (Weighted Relative ENTropy or WRENT score) in uORF-containing transcripts for their agreement with the sequence motifs. While these weighted PSSMs qualitatively resemble the unweighted PSSMs typically used to define sequence motifs (such as the Kozak consensus sequence for translation initiation; Supplementary Fig. 1a,b), weighting for TE quantitatively improved the correlation between relative entropy scores and TEs (Supplementary Fig. 1c,d). A similarly-constructed, weighted initiation context PSSM for uORFs in

transcripts with one non-overlapping uORF did not produce a motif with significant information content (Fig. 1a, inset). These results indicate that in contrast to CDSes, uORFs do not have distinct initiation sequence contexts that promote their translation.

To further compare the initiation contexts of uORFs and CDSes, we used the CDS-derived weighted PSSM (Fig. 1a). Initiation context WRENT scores at uORFs varied widely, but were generally unfavourable for translation initiation (Fig. 1b): only ∼17% of uORFs had more favourable initiation contexts than the median initiation context of CDSes. In addition, transcripts with fewer uORFs tended to have less favourable uORF WRENT scores and more favourable CDS WRENT scores (Supplementary Fig. 1e), which is consistent with the efficient CDS translation of transcripts with fewer uORFs[8,23]. These results provide additional evidence that uORF initiation contexts in general have been under selective pressure to be weakly translated.

**uORF initiation context secondary structure.** RNA secondary structure throughout the transcript may affect translation in multiple ways. RNA secondary structure upstream of ORF starts may impede scanning ribosomal PICs and 60S ribosomal subunit joining, while RNA secondary structure immediately downstream of ORF starts may facilitate start site localization by arresting ribosomal PICs at ORF starts, or impair the start of translation elongation after initiation[2,24].

We characterized the RNA secondary structure around all ORF starts within a transcript by determining ensemble free energy (EFE) profiles. EFE profiles were calculated by the ViennaRNA package[25] in sliding 35-nucleotide windows around all AUG codons (Supplementary Fig. 2a). We found that RNA secondary structure around AUG codons varied significantly between different regions of the transcript, being most stable (that is, lowest EFE scores) around AUGs at CDS and uORF starts, and least stable in 3′ UTRs and within CDSes (Fig. 1c). In addition, CDS starts are characterized by a significant region of increased stability (∼0.3 kcal mol$^{-1}$) immediately downstream of the AUG; in contrast, the region downstream of the AUG start codon of uORFs was less stable (Fig. 1c), an effect that was even more pronounced in transcripts with fewer uORFs (Supplementary Fig. 2b). Notably, these regions of differing stability, which could play a role in translation start site selection, were absent in ORFs beginning inside the CDS and in the 3′ UTR (Fig. 1c). These observations suggest that secondary structure downstream of scanning PICs may promote translation initiation at CDSes by preferentially arresting scanning PICs at CDS but not at uORF starts.

To identify the regions around ORF starts where RNA secondary structure could most affect translation, we correlated the RNA secondary structure EFEs at various positions around CDS starts with their respective CDS TEs (for transcripts lacking uORFs). We found two regions of maximal correlation between RNA secondary structure EFE and CDS TE centered at the 35-nucleotide windows beginning −25 and +1 nucleotides from the CDS start (Supplementary Fig. 2c). Examining the secondary structure EFEs at these two positions for uORFs and CDSes revealed that uORFs and CDSes varied substantially in their initiation context secondary structure (Fig. 1d and Supplementary Fig. 2d). We found that secondary structure stability at uORF starts correlated inversely with the number of uORFs in a transcript (Fig. 1d and Supplementary Fig. 2d-g; see Supplementary Note for a discussion of the interconnected relationships amongst uORF and 5′ leader secondary structure, 5′ leader GC content, and number of uORFs).

**uORF position with respect to CDS**. Following uORF translation, post-termination 40S ribosomal subunits may remain attached and continue scanning to reinitiate at downstream CDSes[3,26]. The efficiency of reinitiation has been observed to decrease as the distance between uORFs and CDSes decreases[22]. To characterize the potential of uORFs to allow reinitiation at downstream CDSes, we examined the positional distribution of uORFs in vertebrate 5′ leaders. Depletion in the distribution of AUG codons had been previously described[27]; we extended these

analyses to uORF ends, which allowed us to consider the effects on the efficiency of reinitiation and uORF repressiveness. While uORFs were found to be broadly distributed in 5′ leaders, uORF starts and uORF ends were depleted near the CDSes (Fig. 1e,f and Supplementary Fig. 3a–c). Although this effect was detected in all three vertebrates, zebrafish transcripts showed the greatest depletion of uORF starts and stops. While the position-specific depletion of AUGs near CDS starts (Fig. 1e) was observed in all three frames (Supplementary Fig. 3d–f), stop codon trinucleotides

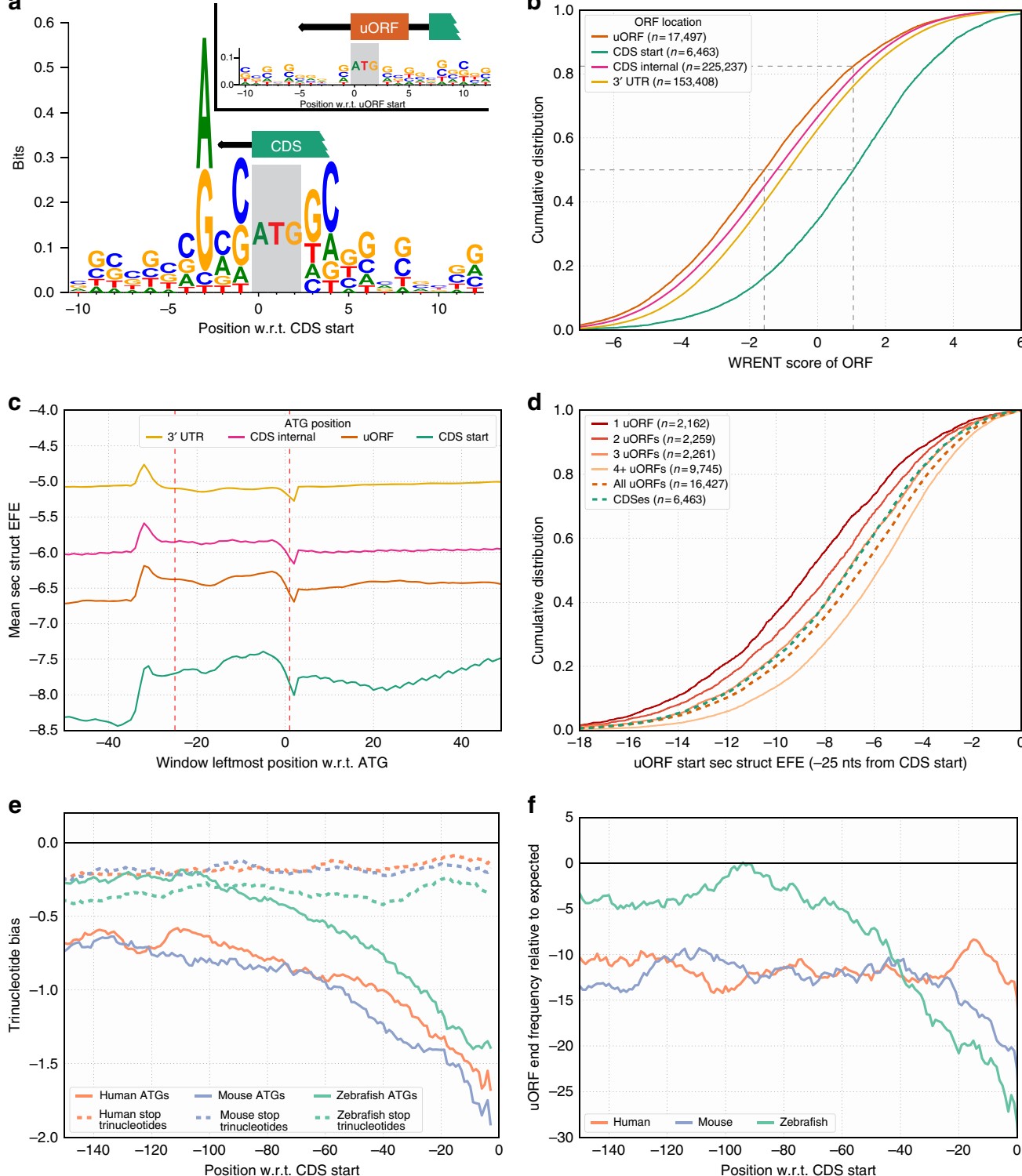

were only depleted near CDS starts if they were preceded by an AUG and thus delineated the end of a uORF (Fig. 1f and Supplementary Fig. 4). These observations indicate that uORFs are depleted in the region ~100 nucleotides upstream of the CDS within vertebrate 5′ leaders, coinciding with the region where uORF positioning diminishes the efficiency of reinitiation[22].

Taken together, our sequence analyses reveal that vertebrate uORFs tend to have features associated with weak repressiveness: they have initiation sequences and secondary structures unfavourable for their translation, and they are depleted from regions closest to the CDS where they would be most repressive.

**uORFs are modestly repressive on average.** The sequence features of uORFs and proteomics data[8] suggest that uORFs are only modestly (~15–30%) repressive for downstream CDS translation. To directly quantify the TE of CDSes (as opposed to inferring it from protein and RNA levels), we calculated the density of ribosome profiling reads over individual CDSes and normalized it by transcript abundance. This approach allowed us to compare the TE of CDSes in mRNAs with varying numbers of uORFs. We observed that the presence of uORFs in 5′ leaders was associated with reduced transcript levels and reduced CDS translation (Supplementary Fig. 5a,b), which together resulted in a decrease in CDS TE (averaging 30–48% reduction amongst the three species; Fig. 2a). Moreover, uORFs were associated with a reduction in CDS TE in a 'dose-dependent' manner: more uORFs in transcripts correlated with increased translation over the 5′ leaders and reduced translation in the CDS (Fig. 2a–c).

Although uORFs can be repressive, studies during yeast meiosis have suggested that they might not explain the majority of gene-to-gene variation in CDS translational efficiency[28]: instead of a negative correlation between uORF and CDS TEs, uORF and CDS translation had been found to be weakly but positively correlated. To determine whether a similar trend holds true in vertebrates, we compared uORF and CDS TEs in the subset of transcripts with only one non-overlapping uORF (Supplementary Fig. 5c). Indeed, we observed a significant and positive correlation between uORF and CDS TEs in all three vertebrate ribosome profiling data sets (Fig. 2d). These observations were further supported by positive correlations of TEs between both uORFs in transcripts with two non-overlapping uORFs (Supplementary Fig. 5d), and the positive correlations of TEs between 5′ leaders and CDSes in transcripts with varying numbers of uORFs (Fig. 2c). These observations suggest that at least in the biological samples represented by the ribosome profiling data sets, other forms of translational regulation, such as recruitment of the 43S PIC to the 5′ cap[29], are dominant in specifying the efficiency of CDS translation.

**uORF features correlate with translation and repressiveness.** To integrate the above analyses, we asked whether there is a relationship between uORF sequence features and uORF repressiveness. As expected, analyses of transcripts with one non-overlapping uORF revealed that more favourable initiation context sequences and less-stable secondary structures correlate with increased uORF TE (Fig. 3a,b, Supplementary Fig. 6a,b and Supplementary Table 1), while uORF TE is independent of the distances between uORFs and downstream CDSes (Fig. 3c, Supplementary Fig. 6c and Supplementary Table 1).

To estimate the repressive effects of uORFs on CDS translation, we calculated the ratio between uORF and CDS TEs and correlated it with uORF sequence features, reasoning that translation of a uORF would occur at the expense of translation of the downstream CDS. We found that each individual uORF sequence feature correlated significantly with uORF repressiveness: more favourable initiation context sequences, less-stable initiation context secondary structure, and reduced distance from the CDS correlated with increased uORF repressiveness (Fig. 3a–c and Supplementary Fig. 6d–f). While more favourable initiation context sequences and reduced distance from the CDS correlated with reduced CDS TE (Fig. 3a,c and Supplementary Fig. 6g,i), the opposite effect was observed for the secondary structure around uORF starts: less-stable secondary structure at uORF starts correlated with increased CDS TE (Fig. 3b; Supplementary Fig. 6h; Supplementary Table 1), suggesting that CDS translation is more affected by features that directly impede 43S PIC scanning over 5′ leaders than by impeding uORF translation initiation; further analysis is presented in the next section.

**uORF repressiveness is specified by transcript features.** While the above analyses revealed that various uORF sequence features individually correlate with uORF repressiveness, it was still unclear whether their contributions to uORF repressiveness were independent of each other. To determine how various sequence features (including that of 5′ leaders and CDSes) collectively specified uORF repressiveness, we constructed linear models with different sets of sequence features. For 5′ leaders, we considered their mean secondary structure EFEs and lengths; for CDSes, we considered their WRENT scores, their secondary structure EFE 5′ and 3′ of the starts, and their mean secondary structure EFE).

Linear modelling of uORF repressiveness (Supplementary Fig. 7a,b) with only uORF sequence features (in transcripts with one non-overlapping uORF) confirmed that they contributed largely in an additive manner, together accounting for ~4.1-fold variation in uORF repressiveness (Table 1; Supplementary Fig. 7a,b; a conservative estimate derived from 95% of the endogenous variation of sequence features). While the inclusion

---

**Figure 1 | uORF sequence features are associated with weak repressiveness.** (**a,b**) Analysis of initiation context sequence in mouse. (**a**) Sequence motif over CDS starts (±10 nucleotides around the annotated AUG start) constructed from CDS TE-weighted position-specific scoring matrices (PSSMs) of coding transcripts lacking uORFs. Height on vertical axis represents weighted relative entropy (WRENT) at individual positions. Inset shows sequence motif at uORF starts, using uORF TE-weighted PSSMs of coding transcripts with one non-overlapping uORF. (**b**) Cumulative distribution of WRENT scores around AUGs at various positions in coding transcripts. Dotted lines indicate median uORF and CDS WRENT scores, as well as the proportion of uORFs (~83%) with WRENT scores less than the median CDS WRENT score. (**c,d**) Analysis of initiation context secondary structure in mouse. (**c**) Meta-profiles of predicted secondary structure ensemble free energies (EFEs; sliding 35-nucleotide window) around AUGs in 5′ leaders, CDSes, and 3′ UTRs. A more negative EFE indicates more stable secondary structure. Red dotted lines indicate the positions −25 and +1 from ORF start that were used for subsequent analyses. (**d**) Cumulative distribution plot of initiation context secondary structure (at position +1 from the ORF start) of uORFs in transcripts with varying number of uORFs. Distributions of secondary structure EFEs for all uORFs and for CDS initiation contexts are indicated (dashed lines). (**e,f**) Distribution of uORFs. (**e**) ATG (that is, start codon; solid lines) and stop codon (dashed lines) moving average of positional trinucleotide biases in 5′ leaders are plotted against their position with respect to the CDS start, for all 3 vertebrates. ATGs, but not stop codon trinucleotides are specifically depleted near the CDS start. (**f**) Depletion in the frequency of uORF ends (moving average over 24 nucleotides) relative to expected frequencies from shuffled 5′ leaders, plotted against uORF-end position with respect to CDS, for all 3 vertebrates. uORF ends are specifically depleted in the 5′ leader near the CDS, most significantly in zebrafish and mouse transcripts, less so in human transcripts.

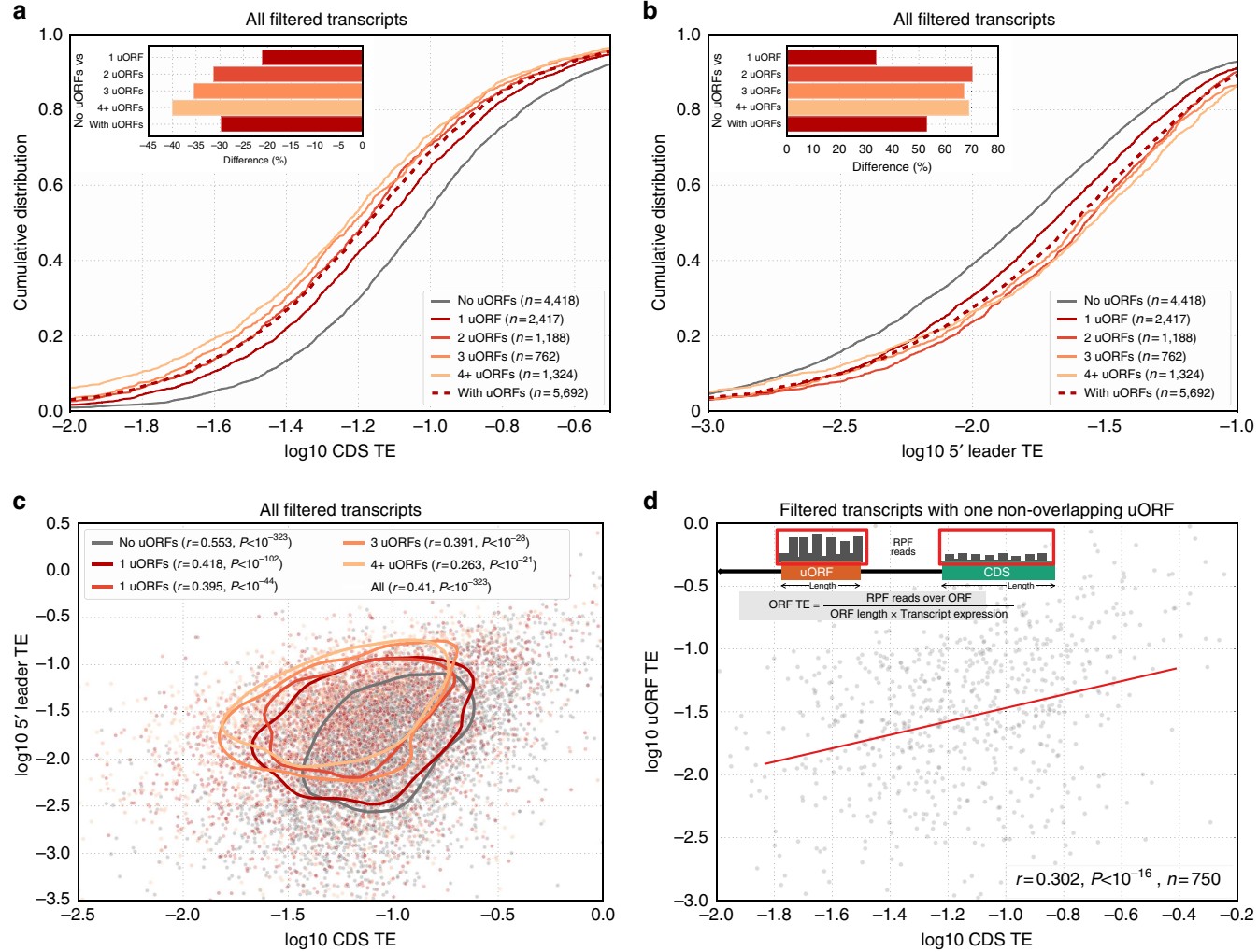

**Figure 2 | uORFs are modestly repressive on an average. (a)** Cumulative distribution of CDS TEs in transcripts grouped by their number of uORFs. The presence of uORFs is associated with a reduction of CDS TEs (inset; between 21 and 40% reduction with increasing number of uORFs, averaging 30%). **(b)** Cumulative distribution of 5′ leader TEs in transcripts grouped by their number of uORFs. The presence of uORFs is associated with an increase in 5′ leader TEs (inset; between 33 and 71% increase with increasing number of uORFs, averaging 53%). **(c)** Relationship between 5′ leader and CDS TE in transcripts with varying numbers of uORFs. While having more uORFs is associated with increased 5′ leader TE and reduced CDS TE, overall, 5′ leader and CDS TEs correlate with each other ($r = 0.41$, $P < 10^{-323}$). Contours indicate the 20th percentile values of a bivariate Gaussian kernal density estimator for each subset of transcripts. **(d)** uORF TEs correlate with cognate CDS TEs for transcripts with one non-overlapping uORF. Red line indicates ridge regression linear fit. Transcript schematic outlines how TEs of individual ORFs are calculated. uORF TEs correlate weakly, but significantly and positively with cognate CDS TEs ($r = 0.302$, $P < 10^{-16}$).

of CDS sequence features could explain some additional variation in uORF repressiveness (Fig. 3d,e; totaling ∼5.6-fold variation), adding 5′ leader sequence features in the linear modelling did not add predictive power for single uORF transcripts (see PRESS or predicted residual sum of squares values in Table 1). In addition, linear modelling suggested that the mean secondary structure over entire CDSes, but not specifically at CDS starts, accounted for the bulk of the contribution of CDS secondary structure to uORF repressiveness (Fig. 3e). Our analyses show that the combination of various features over the entire transcript contribute towards uORF repressiveness.

**uORF and 5′ leader features contribute to CDS TE.** The positive correlation of uORF initiation context secondary structure EFE with both uORF repressiveness and CDS TE (Fig. 3b) suggests that uORF sequence features may act more directly to modulate CDS TE, rather than indirectly by modulating uORF

translation. To dissect these relative contributions, we quantified the contributions of various sequence features toward CDS translation by constructing linear models of CDS TE with different combinations of uORF, 5′ leader and CDS sequence feature sets.

Analysis of linear models for transcripts with only one non-overlapping uORF revealed that CDS TE is specified by a combination of uORF, CDS and 5′ leader sequence features (Fig. 3f,g; Table 1; Supplementary Fig. 7c–h). Among uORF sequence features, both uORF lengths and the distances between uORFs and CDSes contributed significantly towards specifying CDS TE (Fig. 3g; the seemingly counter-intuitive positive association between 5′ leader lengths and CDS TE is further discussed in the Supplementary Note). Contributions by uORF and CDS WRENT scores were similar in magnitude, but in the opposite directions (Fig. 3g). With respect to secondary structure, our modelling revealed that the mean secondary structure stabilities of entire 5′ leaders and CDSes, but not specifically at

uORF and CDS starts, were associated with reduced CDS TE (Fig. 3g).

To model additional features such as the density of uORFs on 5′ leaders, we expanded our analyses to transcriptome subsets with varying numbers of uORFs (all transcripts, transcripts with uORFs and transcripts without uORFs; Supplementary Fig. 8; Supplementary Tables 2 and 3). This allowed us to more rigorously measure the dose-dependent effects of uORFs on CDS TE (Fig. 2a) by jointly considering other transcript sequence features. Our analyses confirmed that uORFs have a generally repressive and dose-dependent effect on CDS TE (uORF density in 5′ leader is negatively associated with CDS TE; Supplementary Fig. 8b,e). Altogether, the sequence features we examined at 5′ leaders, uORFs and CDSes accounted for ∼4.3-fold variation in

CDS TE (estimated from 95% of the endogenous variation of sequence features; Fig. 3f; Table 1). Consistent with observed the positive correlation of uORF TE and CDS TE (Fig. 2d), uORF sequence features contribute less than 5′ leader and CDS sequence features towards specifying CDS TE (Fig. 3g), suggesting that features other than uORFs are the primary determinants of CDS TE.

**Conservation of 5′ leader and CDS translation.** Transcript expression had previously been found to be broadly conserved among vertebrates[30]. To determine whether translation over 5′ leaders and CDSes is similarly conserved, we compiled lists of orthologous transcripts in human, mouse and zebrafish, and compared their sequence features and translation over 5′ leaders

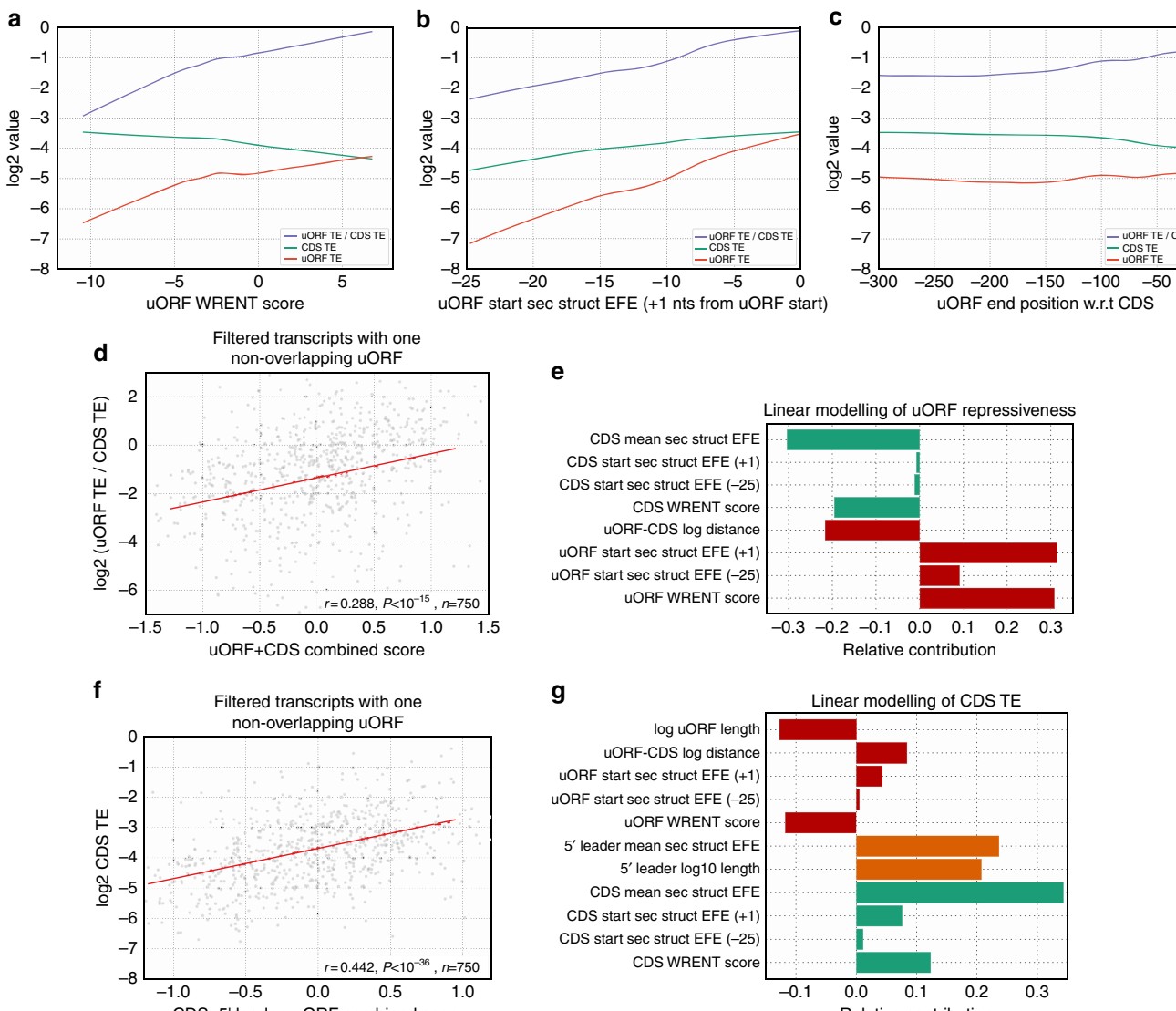

**Figure 3 | Modelling of uORF repressiveness and CDS TE with various transcript sequence features.** (**a–c**) Relationship between (**a**) uORF initiation context sequence, (**b**) secondary structure and **c**. uORF-end position w.r.t. (with respect to) CDS against uORF TE (red), CDS TE (green) and uORF TE / CDS TE (blue). uORF repressiveness (estimated from uORF TE / CDS TE) increases with more favourable initiation context sequence (**a**), decreased secondary structure around uORF initiation sites (**b**) and reduced uORF-CDS distance (**c**). All lines are LOWESS fits representing statistically significant rank and linear correlations (P<0.05; details in Supplementary Fig. 6) except for uORF TE against uORF-end position w.r.t. CDS (**c**). (**d,e**) Linear modelling of uORF/CDS TE with uORF and CDS sequence features, for transcripts with one non-overlapping uORF. Scatter plot of uORF TE / CDS TE against a combined score that integrates uORF and CDS sequence features; red line indicates the ridge regression linear fit (**d**); relative contributions of individual sequence features to the combined score are depicted in **e**. (**f,g**) Linear modelling of CDS TE with uORF, 5′ leader and CDS sequence features, for transcripts with one non-overlapping uORF. Scatter plot of CDS TE is against a combined score that integrates various sequence features; red line indicates the ridge regression linear fit (**f**); relative contributions of individual sequence features to each score are depicted in **g**.

and CDSes. We found that CDS translation and TE are broadly conserved (Fig. 4a; Supplementary Fig. 9a); in fact, the conservation of CDS translation (measured as the correlation of CDS ribosome profiling read densities) is greater than the conservation of transcript expression ($r = 0.727$ versus $r = 0.6$; Supplementary Fig. 9b; Supplementary Table 4), suggesting that translational regulation contributes additively to the conservation of gene expression. Correspondingly, the divergence of CDS translation between species (that is, the difference between Z-normalized CDS ribosome profiling read densities from orthologous transcripts) is well predicted by the differences in both transcript expression and CDS TE, with CDS TE contributing at least half as much as transcript expression (Supplementary Fig. 10).

To assess the conservation of uORF-mediated translational regulation, we compared the densities of ribosome profiling reads over entire 5′ leaders of orthologous transcripts instead of at individual uORFs because unambiguous assignment of ortholo-gous uORFs is unfeasible given their short sequence lengths, particularly in transcripts with multiple uORFs. We found that 5′ leader TEs, ribosome profiling read densities, and the ratios of 5′ leader to CDS TEs ('5′ leader repressiveness', analogous to our measure of uORF repressiveness) show a positive correlation between species (Fig. 4b; Supplementary Fig. 9c,d; Supplementary Table 4), even when the number of uORFs and the length of the 5′ leader differ (Supplementary Fig. 11 and Supplementary Tables 5–7). These findings suggest that the overall repressiveness of the 5′ leaders is also broadly conserved, and may thus contribute to the conservation of CDS TE.

We observed that the number of uORFs tended to be similar between orthologous transcripts, correlating negatively with CDS TE and translation (Fig. 4a and Supplementary Fig. 9a), and positively with 5′ leader repressiveness (Fig. 4b). Other transcript sequence features in 5′ leaders (Fig. 4c,d) and CDSes (Fig. 4e,f) are also conserved, albeit to varying degrees. We observed that more conserved sequence features tended to have stronger contributions towards specifying CDS TE, for example, CDS mean secondary structure EFE is highly conserved (Fig. 4f), and also contributes substantially towards specifying CDS TE (Fig. 3g; Supplementary Fig. 8b,e,h). Similarly, the divergence of sequence features between species is also correlated with the divergence of 5′ leader repressiveness (Supplementary Fig. 12a–f) and CDS TE (Supplementary Fig. 12g–l). These findings suggest that features such as the presence of uORFs, as well as the stability of secondary structure within 5′ leaders and CDSes contribute to the evolutionary variation in 5′ leader and CDS translation between species.

## Discussion
Our study reveals the wide range of uORF-mediated translational repression in vertebrates and provides four major insights: first,

uORFs are generally modestly repressive towards downstream CDS translation; second, uORF repressiveness and CDS TE is modulated by various sequence features; third, genomic variation in uORF repressiveness contributes less than other transcript features towards specifying CDS TE; fourth, the repressiveness and sequence features of uORFs and 5′ leaders are broadly conserved.

Our work builds on and extends previous studies that analysed the roles of uORFs in translational regulation[8,28,31–33], as well as studies that looked more broadly at sequence features that affect translation[27,34,35]. In particular, we examined the contribution of uORFs genome-wide towards specifying the level of CDS translation. Apart from providing a global view of uORF-mediated translational repression, our approach allowed us to characterize the existing endogenous variation amongst uORFs. We found that while uORF sequence features generally disfavour uORF translation (Fig. 1), thus making them less repressive (Fig. 3a–c), uORFs still contribute significantly and negatively towards CDS TE at a genome-wide scale (Fig. 2a; Supplementary Fig. 8b,e).

Linear modelling of various sequence features in hundreds to thousands of transcripts enabled us to dissect the contributions of various transcript features to uORF repressiveness and CDS TE (Fig. 3d,e). We found that uORF sequence features such as the nucleotide sequence around uORF starts, the distance of the uORF from the CDS, and the number of uORFs within a transcript all contribute to uORF repressiveness. These sequence features can be as important for specifying CDS TE as CDS sequence features such as the nucleotide sequence surrounding the CDS start (also known as the Kozak initiation context; Fig. 3g). However, consistent with uORFs being overall a minor determinant of CDS TE (Fig. 2d), we find that some transcript features at 5′ leaders and CDSes (such as the mean secondary structure over their entire length) have a greater influence on CDS TE than uORF sequence features (Fig. 3g). These transcript features at 5′ leaders and CDSes also tend to be significantly conserved over evolution (Fig. 4c-f); consistent with being important for specifying CDS TE, when these transcript features do differ between orthologous vertebrate transcripts, the differences can explain some of the corresponding differences in CDS TE (Supplementary Fig. 12).

While the average repressiveness of uORFs may be modest, a subset of transcripts are substantially modulated in their expression. Moreover, modest but widespread alterations in translation have previously been shown to have significant biological consequences[36,37], although it is still unclear whether the resultant phenotypes are primarily due to changes to just a handful of genes. As our analyses were done on data sets from a limited number of cell types, it is possible that uORFs could have more substantial average effects on translation in other cell types

**Table 1 | Summary statistics for the linear modelling of uORF repressiveness and CDS TE using various sequence feature sets.**

| Parameter modelled | Sequence feature set | Pearson r | P value | Fold change | PRESS | RESS | n |
|---|---|---|---|---|---|---|---|
| uORF repressiveness | uORF | 0.2306 | $1.64 \times 10^{-10}$ | 4.11 | 3,682 | 3,643 | 750 |
| | uORF + 5′ leader | 0.2371 | $4.79 \times 10^{-11}$ | 4.45 | 3,689 | 3,632 | 750 |
| | uORF + CDS | 0.288 | $8.71 \times 10^{-16}$ | 5.609 | 3,602 | 3,529 | 750 |
| | uORF + CDS + 5′ leader | 0.2922 | $3.14 \times 10^{-16}$ | 5.979 | 3,610 | 3,519 | 750 |
| CDS TE | CDS | 0.3468 | $1.29 \times 10^{-22}$ | 3.175 | 1,103 | 1,089 | 750 |
| | CDS + uORF | 0.421 | $1.42 \times 10^{-33}$ | 4.227 | 1,047 | 1,019 | 750 |
| | CDS + 5′ leader | 0.4148 | $1.50 \times 10^{-32}$ | 4.099 | 1,043 | 1,025 | 750 |
| | CDS + 5′ leader + uORF | 0.4421 | $3.05 \times 10^{-37}$ | 4.337 | 1,029 | 996 | 750 |

CDS, coding sequence; TE, translation efficiency
Summary statistics includes the linear correlation and prediction errors (using the predicted residual sum of squares or PRESS statistic), as well as the fold change in uORF repressiveness observed over a 95% range in the combined scores.

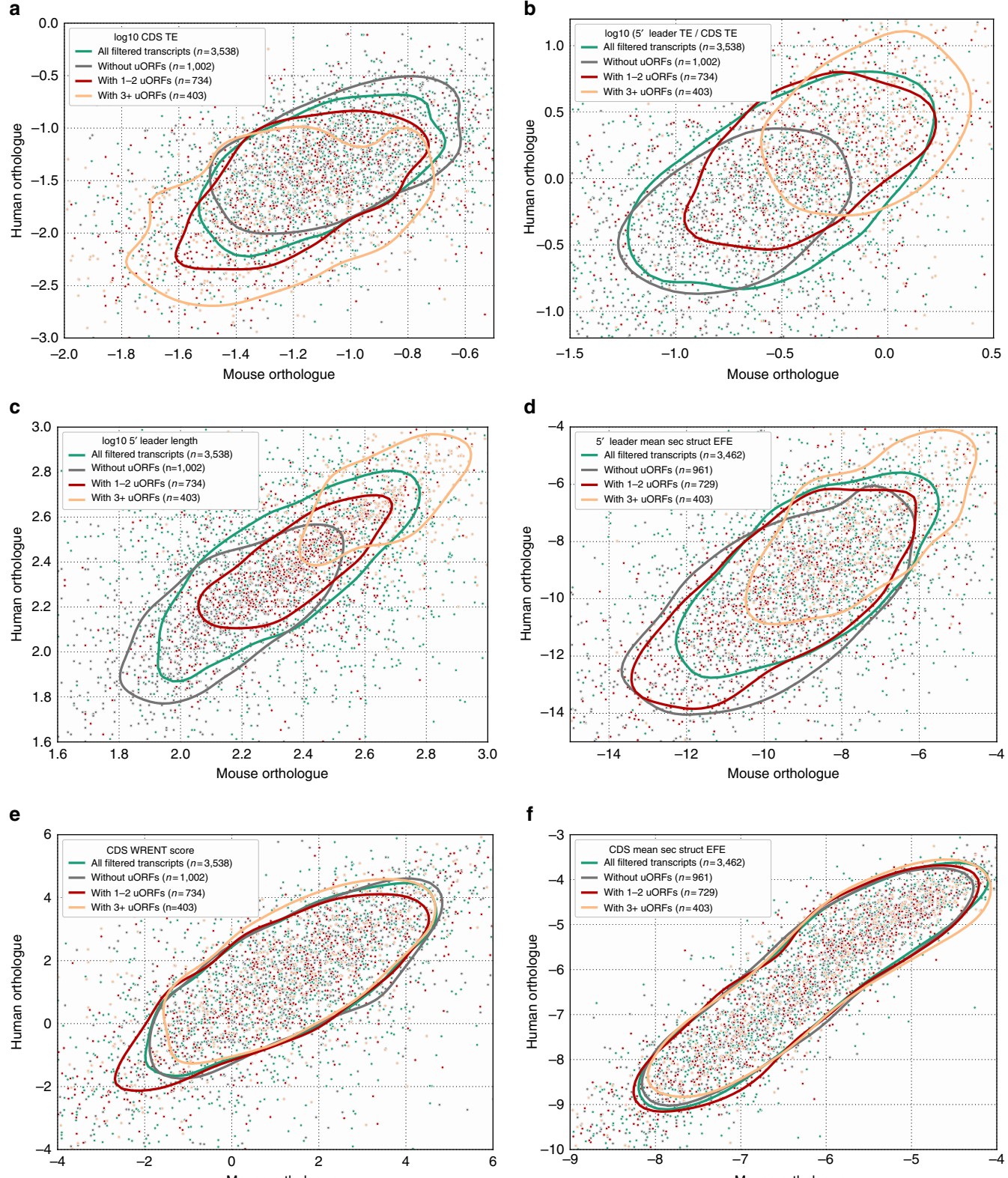

**Figure 4 | Translation and sequences features in 5′ leaders and CDSes are conserved between mouse and human.** (**a–f**) Scatter plots of CDS TE (**a**), the ratio of 5′ leader TE to CDS TE (**b**), 5′ leader length (**c**), 5′ leader mean secondary structure EFE (**d**), CDS WRENT score (**e**) and CDS mean secondary structure EFE (**f**), for human and mouse orthologous transcripts. Data points are coloured by the number of uORFs in the orthologous pairs of transcripts, while contours (20th percentile values of a bivariate Gaussian kernal density estimator) depict the distribution of each subset of points. Translation and sequence features in orthologous transcripts are generally well correlated between human and mouse, which is indicative of their conservation; conservation of translation (**a,b**) and 5′ leader sequence features (**c,d**) additionally co-vary with the number of uORFs in a transcript, while conservation of CDS sequence features is largely independent of uORF number (**e,f**).

or under various stresses[10,38–43], where specific trans-acting factors may alter the translation of uORFs in sequence specific ways. Our analytical methods could be applied to other systems to gain further quantitative insight into how various parameters of 5′ leader and CDS translation vary in different contexts, for example, whether the effects of 5′ leader secondary structures stability on downstream CDS translation differ when various complements of RNA helicases are present.

It has been shown that cells are capable of expressing proteins at the right levels and in the right ratios[44] over a dynamic range spanning six orders of magnitude[45]. Our discovery that 5′ leader repressiveness is conserved amongst vertebrates (Fig. 4b) supports the notion that their translation could contribute to the precise tuning of protein levels. Although transcriptional control accounts for the majority of variation in gene expression[46,47], translational tuning via uORFs may further refine expression levels to their optimum, while providing opportunities for additional layers of post-transcriptional regulation.

Finally, the finding that differences in uORF sequence features correlate with differences in uORF repressiveness in orthologous genes between species raises the possibility that sequence variation at individual genes may contribute to expression level and phenotypic diversity within a species. In humans, sequence variations in 5′ leaders have been statistically associated with variation in gene expression[48], while mutations at uORFs have been shown to contribute to disease[49]. Our methods of analysing the effect of various transcript sequence features on downstream translation may be used to explore the impact of non-coding sequence variation within transcripts and provide a molecular framework to understanding their effects on gene expression and physiology.

## Methods

**Software and code availability.** All data were analysed via a combination of existing software (Tuxedo suite tools[50] for short-read alignment and quantification; ViennaRNAfold[25] for secondary structure prediction), custom shell and Python scripts, and existing Python libraries. These analyses, along with the underlying code, are fully documented in iPython Notebooks[51] as Supplementary Software; the most recent versions can be found at http://github.com/chewgl/uORF_repressiveness_supplemental/. A brief description of key data sources and methods follow:

**Ribosome profiling and matched RNA-Seq data.** Ribosome profiling and matched RNA-Seq data analysed in this study had been previously published: the mouse data from mouse embryonic stem cells[10]; human data from mitotic HeLa cells[11]; and zebrafish data from shield stage embryos[9].

**Gene annotations and mapping.** Ribosome profiling and RNA-Seq data were mapped[21] to GRCh37/hg19, GRCm38/mm10 and Zv9 assemblies of the human, mouse and zebrafish genomes, respectively, using gene annotations based on Ensembl Release 70, as compiled in Illumina's iGenomes collection. Only one transcript per gene (as collated by UCSC's gene-transcript-protein tables) was analysed: if there were multiple annotated transcripts per gene, then only the transcript with the longest CDS, and then the longest 5′ UTR was used. Orthologous transcripts were determined from the list of high-confidence one-to-one orthologous genes in Ensembl Release 75. Ribosome profiling data were reduced to single-nucleotide P-site alignments[9,10].

**Open reading frames.** ORFs in all three species were defined based on sequence, beginning with an AUG codon and ending with an in-frame stop codon (Supplementary Fig. 13a). No minimum length was required, reasoning that an initiating ribosome does not 'know' how long an ORF will be. Thus, all AUGs in 5′ leaders were considered as potential uORF starts: for example, two in-frame AUGs were considered as two separate uORFs that ended at the same stop codon. When considering transcripts with only one uORF, the uORF initiation site was therefore unambiguous. Minimum uORF and CDS lengths of 21 and 100 nucleotides, respectively (Supplementary Fig. 13b) were required only when translation of a uORF was quantified, so as to reduce stochastic noise; in addition, the last 10 nucleotides of an ORF were omitted when calculating TE due to the peak of ribosome profiling reads at ORF stops (Supplementary Fig.13c–e). While we acknowledge the occurrence of non-AUG initiation of translation, especially in 5′ leaders[10,12], it is relatively infrequent compared to canonical AUG initiation[52] and

was therefore not considered. The density of uORFs was calculated to be the number of uORFs (as defined above, equivalent to the number of AUGs) within the 5′ leader, normalized by the length of the 5′ leader.

**Quantification of translational efficiencies and uORF repressiveness.** ORF TEs were calculated by normalizing the density of ribosome profiling reads over the ORF (omitting the last 10 nucleotides) by the associated transcript's FPKM value (as determined by Cufflinks). Repressiveness of uORFs was calculated as the ratio of a uORF's TE to the cognate CDS's TE; similarly, the repressiveness of a 5′ leader was calculated as the ratio of the density of ribosome profiling reads over the 5′ leader to that over the CDS (omitting the last 10 nucleotides of the CDS).

**Initiation context primary sequence.** For analyses of the effect of primary sequence on translation initiation, the initiation sequence context was defined as the 20 nucleotides (10 upstream and 10 downstream) surrounding the start (AUG) codon. To construct a position-specific scoring matrix (PSSM) representing favourable initiation contexts, initiation contexts of CDSes of transcripts lacking uORFs were compiled, with contributions weighted by their translational efficiencies (weights were $\log(1 + TE)$ for non-negative weights). Motifs representing this PSSM were created using Weblogo 3's Python libraries[53]. Other initiation contexts (around AUGs elsewhere in transcripts) were scored using the log-likelihood transform of the constructed PSSM, using the nucleotide frequency of the entire PSSM as the background.

**Secondary structure.** For analyses of the effect of secondary structure on translation initiation, ViennaRNA RNAfold[25] was used to calculate the ensemble free energies (EFEs) over transcripts in sliding 35 (or 25, 30 and 40; Supplementary Fig. 2a) nucleotide windows (1 nucleotide steps). Mean secondary structure was calculated as the mean EFE value of the 35-nucleotide windows within the 5′ leader or CDS; transcripts that contained undefined nucleotides ('N's) in their annotated 5′ leaders and CDSes were omitted. Secondary structure EFEs at positions −25 and +1 with respect to the AUG were chosen for further analyses due to their locally maximal correlation with CDS TE (Supplementary Fig. 2c).

**uORF positional frequencies and biases.** Codon and nucleotide frequencies were determined for each position within the 5′ leader with respect to the CDS start. Trinucleotide bias was calculated as the observed codon frequency at a given position, normalized by the position's expected codon frequency (arising from the background nucleotide frequencies at that position). While this bias is not a direct measure of depletion of trinucleotides (as it is normalized to the background single-nucleotide frequencies), it is a conservative underestimate of depletion. The observed frequency of uORFs ending at positions upstream of the CDS start was normalized for the frequency of uORF ends on shuffled 5′ leader sequences (each leader was shuffled a thousand times, yielding an expected frequency of shuffled sequences). This ratio was plotted with respect to the position from the CDS start.

**Linear modelling.** Linear fitting, modelling and model evaluation was performed using the scikit-learn package[54]. Linear fitting and modelling was performed using ridge regression of Z-scored sequence features (normalized by their endogenous variation). Combined scores were calculated from the sum of Z-normalized sequence features, weighted by their individual coefficients from the linear modelling. Model evaluation used leave-one-out cross validation to calculate predicted residual sum of squares (PRESS) statistics. A small pseudocount of 0.1 was used when modelling the density of uORFs in 5′ leaders for transcripts lacking uORFs.

**Linear modelling of divergence.** For the pairwise comparisons between species, a subset of transcripts with orthologues in both species was defined (using high-confidence one-to-one orthologues, as annotated by Ensembl release 75). Log values of transcript levels (measured in FPKM from RNA-Seq data), CDS ribosome profiling (RP) read density, CDS TEs, and 5′ leader TEs were Z-normalized within these subsets of transcripts. Divergences were calculated as the relative difference between the Z-normalized values between species, which may be positive or negative depending on which orthologue had the larger Z-normalized score. Linear models for the divergence of CDS RP read density were constructed using the divergences of CDS TE, 5′ leader TE and transcript expression (similar to the linear modelling above).

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

## Acknowledgements

We thank E. Valen for technical advice, and A. Murray, C. Hunter, S. Ramanathan, R. Losick, A. Subramaniam, D. Koppstein, Y.Q.S. Soh, S.E. Calvo, M. Rabani, J. Farrell and members of the Schier lab for helpful discussions. Many computations in this paper were run on the Odyssey cluster supported by the FAS Division of Science, Research Computing Group at Harvard University. This research was supported by funding from the Howard Hughes Medical Institute (International Student Fellowship, G.-L.C), the Human Frontier Science Program (HFSP) (A.P.) and the National Institutes of Health (K99 HD076935 (A.P.); R01 HD076708 and R01 GM056211 (A.F.S.)).

## Author contributions

G.-L.C. and A.F.S. conceived the project; G.-L.C. designed and performed all computational analyses with input from A.P. and A.F.S.; all authors wrote the manuscript.

## Additional information

**Competing financial interests:** The authors declare no competing financial interests.

**How to cite this article**: Chew, G.-L. *et al.* Conservation of uORF repressiveness and sequence features in mouse, human and zebrafish. *Nat. Commun.* 7:11663 doi: 10.1038/ncomms11663 (2016).

