## [Peer Review File · Nature Communications]

Reviewer #2 (Remarks to the Author):

In this revised manuscript, which is now being considered for publication in Nature Communications, Chew et al. clarify many of the concerns that I and the other reviewers had in the initial submission. In particular, the increased clarity in explaining how the WRENT score was calculated and how it provides some (albeit a small amount of) additional value over other approaches is helpful. Additionally, the statistical analysis is much improved with a more appropriate and quantitative assessment of the effect of different sequence features on TE.

I still feel that the primary novelty of the work (the comparison between species) is diminished by the requirement to compare whole 5' leader sequences between species rather than individual uORFs, meaning that Figure 4 is somehow difficult to put in context with the rest of the paper. Nevertheless, this manuscript is much improved and provides supporting evidence (using a well-documented and annotated genome-wide analysis) of several concepts that have been described previously. Hence, on balance I am happy to support its publication in Nature Communications.

Minor Comments:

1. In my previous review I queried the effect of transcript length in Figure 2D. As the authors note, FPKM does correct for transcript length - my concern was more to do with the observation that there would be considerably more noise (especially for the uORF regions) for ribosome data when shorter regions were being considered (even after correcting for transcript length). Consequently, I was wondering whether the results in Figure 2D would be clearer if you considered only genes with a CDS and a uORF over a prescribed length - I would have thought there would, on average, be a stronger correlation if genes with only longer genes were considered.
2. I might consider changing the title of the manuscript - the conservation between vertebrates no longer feels like the main theme of the work.

Reviewer #3 (Remarks to the Author):

I'm satisfied with the authors' revisions. My main issue with their original submission regarded their use of statistics (as was noted by the other reviewers as well). They incorporated my suggestions (using

cross validation and the PRESS statistic instead of P-values for model selection), so I am satisfied on that front. Other than that, they've addressed all of our other comments, with the exception that I had suggested that they could use reporter constructs to validate their model experimentally. While I do feel those experiments would be valuable I am fine with the paper being published without them.

Reviewer #4 (Remarks to the Author):

The authors addressed the comments adequately. I have no further suggestions.

Reviewer #2 (Response to reviewer):

Regarding reviewer #2's concern about uORF and CDS lengths, to quote

1. In my previous review I queried the effect of transcript length in Figure 2D. As the authors note, FPKM does correct for transcript length - my concern was more to do with the observation that there would be considerably more noise (especially for the uORF regions) for ribosome data when shorter regions were being considered (even after correcting for transcript length). Consequently, I was wondering whether the results in Figure 2D would be clearer if you considered only genes with a CDS and a uORF over a prescribed length - I would have thought there would, on average, be a stronger correlation if genes with only longer genes were considered.

In our data, we had already considered genes with minimum uORF and CDS lengths: at least 21 nucleotides for uORFs, and at least 100 nucleotides for CDSes. We have made changes to the methods section to make this more explicit. Subsets of uORFs with even longer uORFs still show a weak but positive correlation in their TE with CDS TE; the correlation is never negative.

Regarding the title of the manuscript, we have incorporated the editor's suggestion of "uORF repressiveness and sequence features are conserved in human, mouse and zebrafish".